# Socio-demographic factors associated with early antenatal care visits among pregnant women in Malawi: 2004–2016

Wingston Felix Ng'ambi[1]*, Joseph H. Collins[2], Tim Colbourn[2], Tara Mangal[3], Andrew Phillips[2], Fannie Kachale[4], Joseph Mfutso-Bengo[1], Paul Revill[5], Timothy B. Hallett[3]

1 Health Economics and Policy Unit, Kamuzu University of Health Sciences, Lilongwe, Malawi, 2 Institute for Global Health, University College London, London, United Kingdom, 3 MRC Centre for Global Infectious Disease Analysis, Imperial College London, London, United Kingdom, 4 Ministry of Health, Reproductive Health Directorate, Lilongwe, Malawi, 5 Centre for Health Economics, University of York, York, United Kingdom

* wingston.ngambi@gmail.com, wng'ambi@medcol.mw

**Data Availability Statement:** All relevant data are within the paper and its Supporting Information files.

## Abstract

### Introduction

In 2016, the WHO published recommendations increasing the number of recommended antenatal care (ANC) visits per pregnancy from four to eight. Prior to the implementation of this policy, coverage of four ANC visits has been suboptimal in many low-income settings. In this study we explore socio-demographic factors associated with early initiation of first ANC contact and attending at least four ANC visits ("ANC4+") in Malawi using the Malawi Demographic and Health Survey (MDHS) data collected between 2004 and 2016, prior to the implementation of new recommendations.

### Methods

We combined data from the 2004–5, 2010 and 2015–16 MDHS using Stata version 16. Participants included all women surveyed between the ages of 15–49 who had given birth in the five years preceding the survey. We conducted weighted univariate, bivariate and multivariable logistic regression analysis of the effects of each of the predictor variables on the binary endpoint of the woman attending at least four ANC visits and having the first ANC attendance within or before the four months of pregnancy (ANC4+). To determine whether a factor was included in the model, the likelihood ratio test was used with a statistical significance of P< 0.05 as the threshold.

### Results

We evaluated data collected in surveys in 2004/5, 2010 and 2015/6 from 26386 women who had given birth in the five years before being surveyed. The median gestational age, in months, at the time of presenting for the first ANC visit was 5 (inter quartile range: 4–6). The proportion of women initiating ANC4+ increased from 21.3% in 2004–5 to 38.8% in 2015–

**Funding:** Funding for the project was provided by UK Research and Innovation (UKRI) through the GCRF Thanzi la Onse (Health of All) research programme (MR/P028004/1). During the study period, WN, JC, TC, AP, TM, JMB, PR and TBH worked for on the project. The funder had no role in the study design, data collection and analysis, decision to publish, or the presentation of the manuscript.

**Competing interests:** The authors have declared that no competing interests exist.

**Abbreviations:** ANC, Antenatal care; ANC4+, At least four ANC visits prior to six months of first ANC visit; CHAM, Christian Health Association of Malawi; FANC, Focused Antenatal Care; HIC, High Income Countries; LIC, Low Income Countries; MMR, Maternal Mortality Ratio; MDHS, Malawi Demographic and Health Survey; MPHC, Malawi Population and Housing Census; NSO, National Statistical Office; WHO, World Health Organization; SSA, Sub-Saharan Africa.

16. From multivariate analysis, there was increasing trend in ANC4+ from women aged 20–24 years (adjusted odds ratio (aOR) = 1.27, 95%CI:1.05–1.53, P = 0.01) to women aged 45–49 years (aOR = 1.91, 95%CI:1.18–3.09, P = 0.008) compared to those aged 15–19 years. Women from richest socio-economic position ((aOR = 1.32, 95%CI:1.12–1.58, P<0.001) were more likely to demonstrate ANC4+ than those from low socio-economic position. Additionally, women who had completed secondary (aOR = 1.24, 95%CI:1.02–1.51, P = 0.03) and tertiary (aOR = 2.64, 95%CI:1.65–4.22, P<0.001) education were more likely to report having ANC4+ than those with no formal education. Conversely increasing parity was associated with a reduction in likelihood of ANC4+ with women who had previously delivered 2–3 (aOR = 0.74, 95%CI:0.63–0.86, P<0.001), 4–5 (aOR = 0.65, 95%CI:0.53–0.80, P<0.001) or greater than 6 (aOR = 0.61, 95%CI: 0.47–0.79, <0.001) children being less likely to demonstrate ANC4+.

## Conclusion

The proportion of women reporting ANC4+ and of key ANC interventions in Malawi have increased significantly since 2004. However, we found that most women did not access the recommended number of ANC visits in Malawi, prior to the 2016 WHO policy change which may mean that women are less likely to undertake the 2016 WHO recommendation of 8 contacts per pregnancy. Additionally, our results highlighted significant variation in coverage according to key socio-demographic variables which should be considered when devising national strategies to ensure that all women access the appropriate frequency of ANC visits during their pregnancy.

## Introduction

Following the ratification of the Millennium Development Goals in 2000, significant progress has been made in improving maternal and perinatal health internationally, demonstrated through a global 29% reduction in maternal deaths between 2000 and 2015 [1] and a 19% reduction in stillbirths in the same time period [2]. However, low-income countries (LIC) continue to experience disproportionately greater rates of maternal and perinatal mortality when compared to high income-countries (HIC) [1–4]. An estimated two-thirds of all maternal deaths in 2015 occurred in Sub-Saharan Africa, with the region experiencing a maternal mortality ratio (MMR) twice that of HIC and with latest data suggesting that geographic inequalities in maternal health continue to widen [1]. International and intraregional disparities in maternal health outcomes are, in part, attributable to the substantial variation in both coverage and uptake of key maternity services [5]. Antenatal care (ANC), the care of a woman and her foetus from conception until the onset of labour, is one such service in which coverage, especially within the first trimester of pregnancy, is particularly variable. Many LICs reporting higher MMR than the global average have low ANC coverage [6].

Since its inception in 2002, many LICs have adopted the World Health Organization's (WHO) Focused Antenatal Care (FANC) model in which women are recommended to undertake at least four ANC visits during their pregnancy, at around weeks 12, 26, 32 gestation and between 36 and 38 weeks of gestation [7]. Whilst this model involves fewer visits per-pregnancy than models of care employed across HICs, both women's attendance of their initial ANC visit within the first trimester and attending at least four visits (ANC4+) remains very

low across the region of SSA [8]. Despite low uptake of ANC services under the FANC model, the WHO published recommendations in 2016, doubling the previous number of recommended ANC visits, now renamed as 'contacts', to eight within the duration of a woman's pregnancy at 12, 20, 26, 30, 34, 36, 38 and 40 weeks of gestation [9]. These guidelines are supported by evidence from a number of trials which demonstrates that this model of care, more closely resembling contact-schedules employed across HICs, may lead to a reduction in perinatal death and improvements in women's perception of care-quality when compared to the FANC model [10,11].

Malawi is one such country which, despite demonstrating progress in improving maternal outcomes since 2000 [12], continues to report women attend ANC both later and at a lesser frequency than recommended by the FANC model [13]. Pooled data from the Malawi Demographic and Health Surveys (MDHS) collected between 2000 and 2010 showed that only 10% of women accessed ANC within the first trimester and 49% of women achieved ANC4+ under the FANC model [13]. Ensuring both early access to ANC and ANC4+ for women in Malawi is important, as not only does ANC lead to improved maternal [10], newborn [14] and early childhood outcomes [15] but early initiation of ANC is positively associated with women attending both ANC4+ and attending at eight or more ANC visits in other settings [8]. Additionally, access to ANC is associated with improved probability that women will undergo facility-based delivery with the assistance of a skilled birth attendant, a vital service in improving maternal and perinatal health [16].

In this study we explore the social and demographic factors which are associated with women attending fourth contacts with her first visit occurring during or prior to 4 months gestation in Malawi between 2004 and 2016; we define this 'ANC4+'. We have undertaken an analysis of MDHS data which was collected prior to publication of the WHO's 2016 guidelines and adoption of these guidelines by the Malawian government. Whilst ours is not the first study to explore determinants or timeliness of ANC attendance in Malawi [13,17–20], our study is the first to include data collected as part of the 2015–2016 MDHS, the year prior to the implementation of the most recent WHO ANC guidelines. Our study is also the first to explore what factors are associated with both early initiation of ANC and attendance of four or more ANC visits through the use of a combined outcome variable. Therefore, this study is less likely to over-estimate the true proportion of women with ANC4+ visits since mostly the women with at least 4 ANC visits. Most women with at least four ANC visits but with first ANC visit was after five months tended to have pregnancy complications. In additional to socio-demographic factors we also explored the services that were accessed by the women during their ANC visits. Finally, we believe that the results of this study will provide vital insight into potential barriers for early initiation of ANC4+ visits in Malawi and other similar settings, providing key information to guide policy makers, clinicians or programme managers working in maternal and reproductive health.

## Methods

### Study design

We conducted a secondary analysis of the women's questionnaire data collected from three Malawi Demographic and Health Surveys (MDHS) administered between 2004 and 2016 [21–23]. The Women's Questionnaire is one of the four primary DHS survey questionnaires, accompanying the Household, Men's and Biomarker Questionnaires, employed within the data collection process for the MDHS. This questionnaire is used to collect data from female participants on topics such as maternal and child health and healthcare use, contraception and women's socio-economic status in the country of study. All women aged between 15–49 years

are eligible for inclusion and any relevant participants are identified for recruitment via administration of the national Household Questionnaire. Within this study only those respondents who had given birth during in the preceding five years were included in the analysis.

## Sampling procedure

The sampling frame used for the 2010 and 2015–16 MDHS is the frame of the Malawi Population and Housing Census (MPHC) conducted in Malawi in 2008 while the sampling frame for the 2004–5 MDHS was the 1998 MPHC provided by the Malawi National Statistical Office (NSO) [21–23] The sampling frame is a complete list of all census standard enumeration areas (SEAs) created for the 1998 or 2008 MPHC depending on the wave of the MDHS. A SEA is a geographic area that covers an average of 235 households. The sampling frame contains information about the SEA location, type of residence (urban or rural), and the estimated number of residential households.

The MDHS samples were stratified and selected in two stages. Each district was stratified into urban and rural areas; this yielded 56 sampling strata [21–23]. Sample of SEAs were selected independently in each stratum in two stages. Implicit stratification and proportional allocation were achieved at each of the lower administrative levels by sorting the sampling frame within each sampling stratum before sample selection, according to administrative units in different levels, and by using a probability proportional to size selection at the first stage point sampling [21–23].

## Data management

We extracted and combined data from the 2004–5, 2010 and 2015–16 MDHS. We classified the variables as relating to external environment (rural/urban location, survey year, and region of residence), socio-demographics (age of the woman, household wealth index, education, marital status and number of children ever born), knowledge (frequency of listening to radio or watching television)) and enablers (permission to visit health services, money to pay for health services, distance to health facilities, presence of companion, and desire for pregnancy). We also extracted the tests performed during ANC visits (blood, urine and blood pressure) as well as the services that were received (Iron tablets for 90+ days, HIV testing and counselling, sulfadoxine-pyrimethamine (SP)/Fansidar for malaria prophylaxis) amongst the women that had ANC.

The primary outcome was whether or not women had four or more antenatal care visits with a skilled service provider, namely a doctor/medical officer, clinical officer, assistant clinical officer, or nurse/midwife, with the first visit occurring in or prior to the four months of pregnancy. This was the recommended ANC schedule at the time for all the observations in the dataset (i.e., 2004–2016). This analysis included women with their most recent birth within two years preceding each MDHS.

## Statistical analysis

We calculated counts, weighted percentages, weighted odds ratios (OR) and their associated 95% confidence intervals (95%CI). We performed data management and analysis using Stata version 16 (Stata Corp., Texas, USA). The weighting variable from each of the MDHS was divided by 1000000 [24]. We further calculated the equal weights for each sample cluster and divided the average weight for each cluster by three (as data from three survey rounds being used together), as illustrated by Friedman and Jang in 2002 [25]. We conducted weighted univariate, bivariate and multivariable logistic regression analysis of the effects of each of the predictor variables on the binary endpoint of early initiation of ANC4+. Multiple weighted

logistic regression models were used with a forward and back-ward step-wise selection method. To determine whether a factor was included in the model, the likelihood ratio test (LRT) was used with a statistical significance of P< 0.05 as the threshold.

### Ethical considerations

The individual consent was conducted by National Statistical Office (NSO) of Malawi during the DHS 2004/2005, 2010 and 2015/2016. We obtained permission to use this data from the MEASURE DHS. The Malawi DHS datasets were downloaded from https://www.dhsprogram.com/data/available-datasets.cfm. Furthermore, this study was approved by the College of Medicine Research Committee (COMREC) in Blantyre, Malawi (protocol #: P.10/19/2820). As this study used secondary anonymised data, individual informed consent was not required.

## Results

### Characteristics of women included in this study

The characteristics of the women included in the study are shown in Table 1. A total of 26386 women were evaluated between 2004 and 2016. Of these 6012 (23%) were interviewed in 2004/5 MDHS; 10802 (41%) were interviewed in the 2010 MDHS while 9572 (36%) were interviewed in the 2015/16 MDHS. We observed variation in the proportions of women by age group, with an increasing trend from 15 to 29 years and a decreasing trend from 25 years to 49 years (see Table 1). The median age of the respondents was 26 years (interquartile range (IQR): 22–32). Between 2004 and 2016, the highest proportion of women had between 2 and 3 children previously (37%) while the lowest proportion of the women had six or more children (18%). The median number of previous children was 3 (IQR: 2–5). The majority of women had primary education while the minority had tertiary education (see Table 1). Almost 44% (11702 of 26386) were from households of poor socio-economic level and we observed a higher proportion of the households from poor socio-economic position.

Eighty-seven percent of the women were from rural areas while twelve percent were from the urban areas, and more women were interviewed from rural areas (see Table 1). Although 12815 of 26386 listened to radio between 2004 and 2016, we observed a decreasing trend from 65% in 2004/5 to 30% in 2015/16. Overall, the majority of women (24136 of 26386) did not watch a television (TV) for more than once a week but the numbers watching TV increased from 5% in 2004/5 to 10% in 2015/16. Over the 2004 to 2016 surveys, women cited different barriers for them to access ANC and these barriers had different trends. The major barriers were long distance to health facilities (59% of 26386) and lack of money to use in accessing health services (55% of 26386). Across the survey populations, the majority of the women were married while the lowest proportion were widowed (see Table 1).

### Factors associated with early antenatal care of at least four visits

**Distribution of women by number of ANC visits.** The distribution of women by number of ANC visits is shown in Table 2. Of the 26386, 7449 (28%) of the women had attended early initiation of ANC4+. We observed increasing trends in the proportion of women with early initiation of ANC4+ from 21% in 2004/5 to 37% in 2015/16 (P<0.001). The proportion of women with early initiation of ANC4+ decreased with increasing number of children ever born (see Table 2). The women from richest households (35%) had the highest coverage of early initiation of ANC4+ compared to those from the poorest households (25%). The rural women were less likely to demonstrate early initiation of ANC4+ than the urban women (see Table 2).

**Table 1. Characteristics of women interviewed during the Malawi Demographic and Health Surveys conducted between 2004 and 2016.**

| Characteristics | Total | | 2004–5 | | 2010 | | 2015–16 | |
|---|---|---|---|---|---|---|---|---|
| | Number | % | Number | % | Number | % | Number | % |
| **Total** | 26386 | 100.0 | 6012 | 100.0 | 10802 | 100.0 | 9572 | 100.0 |
| **Age group** | | | | | | | | |
| 15–19 | 2671 | 10.0 | 602 | 9.7 | 997 | 9.0 | 1072 | 11.3 |
| 20–24 | 8212 | 31.4 | 2081 | 35.4 | 3118 | 29.4 | 3013 | 31.3 |
| 25–29 | 6542 | 24.9 | 1486 | 25.0 | 2903 | 26.8 | 2153 | 22.8 |
| 30–34 | 4507 | 16.9 | 894 | 14.4 | 1897 | 17.4 | 1716 | 17.9 |
| 35–39 | 2889 | 10.9 | 583 | 9.7 | 1221 | 11.2 | 1085 | 11.3 |
| 40–44 | 1176 | 4.4 | 273 | 4.4 | 491 | 4.6 | 412 | 4.3 |
| 45–49 | 389 | 1.4 | 93 | 1.4 | 175 | 1.6 | 121 | 1.3 |
| **Region** | | | | | | | | |
| North | 4506 | 15.1 | 746 | 12.4 | 1955 | 15.1 | 1805 | 16.7 |
| Centre | 9204 | 38.3 | 2247 | 40.1 | 3679 | 38.3 | 3278 | 37.3 |
| South | 12676 | 46.6 | 3019 | 47.5 | 5168 | 46.6 | 4489 | 46.0 |
| **Number of children ever born** | | | | | | | | |
| 1 | 5787 | 22.2 | 1284 | 21.4 | 2023 | 19.0 | 2480 | 26.2 |
| 2–3 | 9621 | 36.7 | 2212 | 37.3 | 3859 | 36.2 | 3550 | 37.0 |
| 4–5 | 6274 | 23.6 | 1376 | 22.7 | 2713 | 24.8 | 2185 | 22.8 |
| 6+ | 4704 | 17.5 | 1140 | 18.5 | 2207 | 20.1 | 1357 | 14.0 |
| **Education level** | | | | | | | | |
| None | 4274 | 16.1 | 1481 | 24.0 | 1711 | 15.9 | 1082 | 11.5 |
| Primary | 17753 | 67.1 | 3859 | 64.8 | 7517 | 69.0 | 6377 | 66.5 |
| Secondary | 4104 | 15.7 | 659 | 11.0 | 1507 | 14.4 | 1938 | 20.0 |
| Tertiary | 255 | 1.1 | 13 | 0.2 | 67 | 0.7 | 175 | 1.9 |
| **Wealth index quintile** | | | | | | | | |
| Poorest | 5768 | 21.9 | 1160 | 18.7 | 2439 | 22.3 | 2169 | 23.2 |
| Poorer | 5934 | 22.4 | 1392 | 23.2 | 2446 | 22.4 | 2096 | 22.0 |
| Middle | 5670 | 21.3 | 1389 | 23.4 | 2428 | 22.1 | 1853 | 19.1 |
| Richer | 4968 | 18.6 | 1183 | 19.7 | 2030 | 18.6 | 1755 | 18.0 |
| Richest | 4046 | 15.8 | 888 | 14.9 | 1459 | 14.6 | 1699 | 17.7 |
| **Residence** | | | | | | | | |
| Urban | 3240 | 12.9 | 655 | 11.2 | 1037 | 11.1 | 1548 | 16.0 |
| Rural | 23146 | 87.1 | 5357 | 88.8 | 9765 | 88.9 | 8024 | 84.0 |
| **Sources of antenatal care knowledge** | | | | | | | | |
| **Frequency of listening to radio** | | | | | | | | |
| Less than once a week | 13571 | 52.0 | 2146 | 35.4 | 4767 | 44.7 | 6658 | 70.3 |
| At least once a week | 12815 | 48.0 | 3866 | 64.6 | 6035 | 55.3 | 2914 | 29.7 |
| **Frequency of watching television** | | | | | | | | |
| Less than once a week | 24136 | 91.4 | 5732 | 95.5 | 9762 | 90.0 | 8642 | 90.6 |
| At least once a week | 2250 | 8.6 | 280 | 4.5 | 1040 | 10.0 | 930 | 9.4 |
| **Barriers to access antenatal care** | | | | | | | | |
| **Permission to visit health services** | | | | | | | | |
| No problem | 23076 | 87.2 | 5490 | 91.5 | 9524 | 87.9 | 8062 | 83.8 |
| Big problem | 3310 | 12.8 | 522 | 8.5 | 1278 | 12.1 | 1510 | 16.2 |
| **Money to pay for health services** | | | | | | | | |
| No problem | 11893 | 44.6 | 2157 | 35.4 | 4948 | 46.1 | 4788 | 48.6 |
| Big problem | 14493 | 55.4 | 3855 | 64.6 | 5854 | 53.9 | 4784 | 51.4 |

*(Continued)*

**Table 1.** (Continued)

| Characteristics | Total | | 2004–5 | | 2010 | | 2015–16 | |
|---|---|---|---|---|---|---|---|---|
| | Number | % | Number | % | Number | % | Number | % |
| **Distance to health facilities** | | | | | | | | |
| No problem | 10910 | 41.4 | 2224 | 36.7 | 4274 | 40.6 | 4412 | 45.2 |
| Big problem | 15476 | 58.6 | 3788 | 63.3 | 6528 | 59.4 | 5160 | 54.8 |
| **Presence of companion** | | | | | | | | |
| No problem | 18762 | 70.6 | 4411 | 73.7 | 7380 | 67.8 | 6971 | 71.9 |
| Big problem | 7624 | 29.4 | 1601 | 26.3 | 3422 | 32.2 | 2601 | 28.1 |
| **No drugs at health facility** | | | | | | | | |
| No problem | 13473 | 49.7 | 6012 | 100.0 | 4253 | 38.5 | 3208 | 31.4 |
| Big problem | 12913 | 50.3 | 0 | 0.0 | 6549 | 61.5 | 6364 | 68.6 |
| **No female provider** | | | | | | | | |
| No problem | 20981 | 79.3 | 5158 | 86.1 | 8419 | 77.4 | 7404 | 77.2 |
| Big problem | 5405 | 20.7 | 854 | 13.9 | 2383 | 22.6 | 2168 | 22.8 |
| **Marital status** | | | | | | | | |
| Never married | 871 | 3.3 | 138 | 2.2 | 281 | 2.6 | 452 | 4.7 |
| Married | 22522 | 85.4 | 5257 | 87.7 | 9311 | 86.3 | 7954 | 83.1 |
| Widowed | 445 | 1.7 | 113 | 2.0 | 185 | 1.7 | 147 | 1.5 |
| Divorced | 2548 | 9.6 | 504 | 8.1 | 1025 | 9.5 | 1019 | 10.7 |

% = weighted percentage.

**Crude odds ratios of women having early antenatal care of at least four visits.** The crude odds ratios for early initiation of ANC4+ are presented in Table 3. There was increasing trend in the odds of women demonstrating early initiation of ANC4+ from 2004 to 2016 (P<0.001). There was a decreasing trend in the odds in early initiation of ANC4+ with the number of children ever born (Table 3). Watching TV was associated with higher likelihood of ANC4+ (OR = 1.53, 95%CI: 1.31–1.79, P<0.001).

**Adjusted odds ratios of women having early antenatal care of at least four visits.** The adjusted odds ratios for early initiation of ANC4+ are presented in Table 3. The likelihood of women having early initiation of ANC4+ varied by age, survey year, number of children ever born to the woman, education level and wealth index quintile. After adjusting for age, number of children ever born to the woman, education level and wealth index quintile; there was an increasing trend in the likelihood of having early initiation of ANC4+ by year of survey. Women who were wealthier, and more educated, and married were more likely to have had early initiation of ANC4+ (see Table 3). On the other hand, women with more children were less likely to have reported early initiation of ANC4+ visits.

**Services received by antenatal care women.** The services accessed by women that attended ANC in Malawi are shown in Fig 1. The proportion of women that had a blood sample taken for full blood count increased from 32% in 2004/5 to 91% in 2015/16 (Chi-Square P <0.001). Similarly, there was increasing trend in the proportion of women with urine test from 18% in 2004/5 to 32% in 2015/16 (P<0.001). We also observed an increasing trend in the proportion of ANC women that were given at least two doses of Fansidar/SP for malaria prophylaxis from 79% in 2004/5 to 91% in 2015/16 (P<0.001). The percentage of women that had HIV testing at ANC increased from 0% in 2004/5 to 92% in 2010 and 88% in 2015/16. There was an increase in the proportion of women with blood pressure measurement from 73% in

**Table 2. Distribution of women by number of antenatal care visits in Malawi between 2004 and 2016.**

| Characteristics | Total | | | | 2004–5 | | | | 2010 | | | | 2015–16 | | | |
|---|---|---|---|---|---|---|---|---|---|---|---|---|---|---|---|---|
| | <4 ANC | | ANC4+ | | <4 ANC | | ANC4+ | | <4 ANC | | ANC4+ | | <4 ANC | | ANC4+ | |
| | n | % | n | % | n | % | n | % | n | % | n | % | n | % | n | % |
| **Total** | 18937 | 71.9 | 7449 | 28.1 | 4755 | 78.7 | 1257 | 21.3 | 8199 | 76.0 | 2603 | 24.0 | 5983 | 63.2 | 3589 | 36.8 |
| **Age group** | | | | | | | | | | | | | | | | |
| 15–19 | 1931 | 72.6 | 740 | 27.4 | 467 | 76.7 | 135 | 23.3 | 735 | 74.4 | 262 | 25.6 | 729 | 68.9 | 343 | 31.1 |
| 20–24 | 5857 | 71.4 | 2355 | 28.6 | 1613 | 77.2 | 468 | 22.8 | 2348 | 75.3 | 770 | 24.7 | 1896 | 63.3 | 1117 | 36.7 |
| 25–29 | 4678 | 71.7 | 1864 | 28.3 | 1197 | 80.4 | 289 | 19.6 | 2182 | 75.0 | 721 | 25.0 | 1299 | 61.3 | 854 | 38.7 |
| 30–34 | 3244 | 72.2 | 1263 | 27.8 | 723 | 80.1 | 171 | 19.9 | 1472 | 78.0 | 425 | 22.0 | 1049 | 61.9 | 667 | 38.1 |
| 35–39 | 2077 | 71.8 | 812 | 28.2 | 461 | 77.8 | 122 | 22.2 | 938 | 76.3 | 283 | 23.7 | 678 | 63.6 | 407 | 36.4 |
| 40–44 | 877 | 74.7 | 299 | 25.3 | 222 | 82.0 | 51 | 18.0 | 384 | 79.0 | 107 | 21.0 | 271 | 65.0 | 141 | 35.0 |
| 45–49 | 273 | 71.4 | 116 | 28.6 | 72 | 78.2 | 21 | 21.8 | 140 | 82.5 | 35 | 17.5 | 61 | 50.5 | 60 | 49.5 |
| **Region** | | | | | | | | | | | | | | | | |
| North | 3180 | 70.7 | 1326 | 29.3 | 588 | 77.9 | 158 | 22.1 | 1462 | 75.3 | 493 | 24.7 | 1130 | 62.8 | 675 | 37.2 |
| Centre | 6541 | 71.5 | 2663 | 28.5 | 1804 | 80.1 | 443 | 19.9 | 2707 | 74.1 | 972 | 25.9 | 2030 | 63.0 | 1248 | 37.0 |
| South | 9216 | 72.6 | 3460 | 27.4 | 2363 | 77.7 | 656 | 22.3 | 4030 | 77.8 | 1138 | 22.2 | 2823 | 63.5 | 1666 | 36.5 |
| **Number of children ever born** | | | | | | | | | | | | | | | | |
| 1 | 3905 | 67.3 | 1882 | 32.7 | 951 | 73.3 | 333 | 26.7 | 1435 | 70.8 | 588 | 29.2 | 1519 | 61.6 | 961 | 38.4 |
| 2–3 | 6909 | 72.1 | 2712 | 27.9 | 1762 | 79.6 | 450 | 20.4 | 2937 | 76.2 | 922 | 23.8 | 2210 | 63.0 | 1340 | 37.0 |
| 4–5 | 4586 | 73.6 | 1688 | 26.4 | 1128 | 81.3 | 248 | 18.7 | 2091 | 77.5 | 622 | 22.5 | 1367 | 64.2 | 818 | 35.8 |
| 6+ | 3537 | 75.0 | 1167 | 25.0 | 914 | 79.9 | 226 | 20.1 | 1736 | 78.8 | 471 | 21.2 | 887 | 65.0 | 470 | 35.0 |
| **Education level** | | | | | | | | | | | | | | | | |
| None | 3267 | 76.7 | 1007 | 23.3 | 1206 | 81.4 | 275 | 18.6 | 1337 | 79.0 | 374 | 21.0 | 724 | 67.1 | 358 | 32.9 |
| Primary | 12893 | 72.7 | 4860 | 27.3 | 3065 | 79.1 | 794 | 20.9 | 5767 | 76.7 | 1750 | 23.3 | 4061 | 64.3 | 2316 | 35.7 |
| Secondary | 2678 | 65.6 | 1426 | 34.4 | 478 | 71.1 | 181 | 28.9 | 1061 | 70.6 | 446 | 29.4 | 1139 | 59.8 | 799 | 40.2 |
| Tertiary | 99 | 39.9 | 156 | 60.1 | 6 | 32.7 | 7 | 67.3 | 34 | 50.5 | 33 | 49.5 | 59 | 36.0 | 116 | 64.0 |
| **Wealth index quintile** | | | | | | | | | | | | | | | | |
| Poorest | 4323 | 75.1 | 1445 | 24.9 | 954 | 82.2 | 206 | 17.8 | 1939 | 79.8 | 500 | 20.2 | 1430 | 66.5 | 739 | 33.5 |
| Poorer | 4309 | 72.8 | 1625 | 27.2 | 1111 | 79.1 | 281 | 20.9 | 1873 | 77.1 | 573 | 22.9 | 1325 | 63.9 | 771 | 36.1 |
| Middle | 4110 | 72.7 | 1560 | 27.3 | 1103 | 79.1 | 286 | 20.9 | 1820 | 75.3 | 608 | 24.7 | 1187 | 64.6 | 666 | 35.4 |
| Richer | 3563 | 71.7 | 1405 | 28.3 | 925 | 77.9 | 258 | 22.1 | 1533 | 75.0 | 497 | 25.0 | 1105 | 63.7 | 650 | 36.3 |
| Richest | 2632 | 65.4 | 1414 | 34.6 | 662 | 73.9 | 226 | 26.1 | 1034 | 70.9 | 425 | 29.1 | 936 | 55.8 | 763 | 44.2 |
| **Residence** | | | | | | | | | | | | | | | | |
| Urban | 2108 | 67.3 | 1132 | 32.7 | 491 | 74.8 | 164 | 25.2 | 751 | 73.2 | 286 | 26.8 | 866 | 56.6 | 682 | 43.4 |
| Rural | 16829 | 72.8 | 6317 | 27.2 | 4264 | 79.0 | 1093 | 21.0 | 7448 | 76.4 | 2317 | 23.6 | 5117 | 64.4 | 2907 | 35.6 |
| **Sources of antenatal care knowledge** | | | | | | | | | | | | | | | | |
| **Frequency of listening to radio** | | | | | | | | | | | | | | | | |
| Less than once a week | 9740 | 71.9 | 3831 | 28.1 | 1765 | 81.8 | 381 | 18.2 | 3692 | 77.8 | 1075 | 22.2 | 4283 | 64.7 | 2375 | 35.3 |
| At least once a week | 9197 | 71.9 | 3618 | 28.1 | 2990 | 77.0 | 876 | 23.0 | 4507 | 74.6 | 1528 | 25.4 | 1700 | 59.7 | 1214 | 40.3 |
| **Frequency of watching television** | | | | | | | | | | | | | | | | |
| Less than once a week | 17503 | 72.7 | 6633 | 27.3 | 4550 | 78.9 | 1182 | 21.1 | 7444 | 76.5 | 2318 | 23.5 | 5509 | 64.5 | 3133 | 35.5 |
| At least once a week | 1434 | 63.5 | 816 | 36.5 | 205 | 73.0 | 75 | 27.0 | 755 | 72.0 | 285 | 28.0 | 474 | 50.6 | 456 | 49.4 |
| **Barriers to access antenatal care** | | | | | | | | | | | | | | | | |
| **Permission to visit health services** | | | | | | | | | | | | | | | | |
| No problem | 16620 | 72.1 | 6456 | 27.9 | 4357 | 78.9 | 1133 | 21.1 | 7250 | 76.2 | 2274 | 23.8 | 5013 | 62.9 | 3049 | 37.1 |
| Big problem | 2317 | 70.4 | 993 | 29.6 | 398 | 76.7 | 124 | 23.3 | 949 | 74.8 | 329 | 25.2 | 970 | 64.8 | 540 | 35.2 |
| **Money to pay for health services** | | | | | | | | | | | | | | | | |
| No problem | 8303 | 70.2 | 3590 | 29.8 | 1668 | 76.7 | 489 | 23.3 | 3728 | 75.8 | 1220 | 24.2 | 2907 | 61.3 | 1881 | 38.7 |

*(Continued)*

**Table 2.** (Continued)

| Characteristics | Total | | | | 2004–5 | | | | 2010 | | | | 2015–16 | | | |
|---|---|---|---|---|---|---|---|---|---|---|---|---|---|---|---|---|
| | <4 ANC | | ANC4+ | | <4 ANC | | ANC4+ | | <4 ANC | | ANC4+ | | <4 ANC | | ANC4+ | |
| | n | % | n | % | n | % | n | % | n | % | n | % | n | % | n | % |
| Big problem | 10634 | 73.3 | 3859 | 26.7 | 3087 | 79.7 | 768 | 20.3 | 4471 | 76.2 | 1383 | 23.8 | 3076 | 64.9 | 1708 | 35.1 |
| **Distance to health facilities** | | | | | | | | | | | | | | | | |
| No problem | 7674 | 70.6 | 3236 | 29.4 | 1720 | 76.7 | 504 | 23.3 | 3258 | 76.4 | 1016 | 23.6 | 2696 | 61.9 | 1716 | 38.1 |
| Big problem | 11263 | 72.8 | 4213 | 27.2 | 3035 | 79.8 | 753 | 20.2 | 4941 | 75.8 | 1587 | 24.2 | 3287 | 64.3 | 1873 | 35.7 |
| **Presence of companion** | | | | | | | | | | | | | | | | |
| No problem | 13428 | 71.8 | 5334 | 28.2 | 3487 | 78.6 | 924 | 21.4 | 5625 | 76.3 | 1755 | 23.7 | 4316 | 62.9 | 2655 | 37.1 |
| Big problem | 5509 | 72.1 | 2115 | 27.9 | 1268 | 78.7 | 333 | 21.3 | 2574 | 75.4 | 848 | 24.6 | 1667 | 64.0 | 934 | 36.0 |
| **No drugs at health facility** | | | | | | | | | | | | | | | | |
| No problem | 9991 | 74.2 | 3482 | 25.8 | 4755 | 78.7 | 1257 | 21.3 | 3246 | 76.1 | 1007 | 23.9 | 1990 | 62.7 | 1218 | 37.3 |
| Big problem | 8946 | 69.7 | 3967 | 30.3 | 0 | 0.0 | 0 | 100.0 | 4953 | 76.0 | 1596 | 24.0 | 3993 | 63.4 | 2371 | 36.6 |
| **No female provider** | | | | | | | | | | | | | | | | |
| No problem | 15046 | 71.8 | 5935 | 28.2 | 4090 | 78.8 | 1068 | 21.2 | 6391 | 76.1 | 2028 | 23.9 | 4565 | 62.2 | 2839 | 37.8 |
| Big problem | 3891 | 72.3 | 1514 | 27.7 | 665 | 77.8 | 189 | 22.2 | 1808 | 75.8 | 575 | 24.2 | 1418 | 66.4 | 750 | 33.6 |
| **Marital status** | | | | | | | | | | | | | | | | |
| Never married | 629 | 73.7 | 242 | 26.3 | 110 | 80.2 | 28 | 19.8 | 213 | 74.9 | 68 | 25.1 | 306 | 71.1 | 146 | 28.9 |
| Married | 16134 | 71.7 | 6388 | 28.3 | 4146 | 78.4 | 1111 | 21.6 | 7057 | 75.9 | 2254 | 24.1 | 4931 | 62.4 | 3023 | 37.6 |
| Widowed | 330 | 74.1 | 115 | 25.9 | 92 | 80.2 | 21 | 19.8 | 144 | 78.2 | 41 | 21.8 | 94 | 64.1 | 53 | 35.9 |
| Divorced | 1844 | 73.1 | 704 | 26.9 | 407 | 80.6 | 97 | 19.4 | 785 | 77.4 | 240 | 22.6 | 652 | 65.4 | 367 | 34.6 |

<4 ANC = Less than four early antenatal care (ANC) visits.

ANC4+ = At least four ANC visits prior to six months of first ANC visit.

% = weighted percentage.

2004/5 to 82% in 2015/16. We observed increasing trend in the percentage of women that received iron tablets from 80% to 92% (see Fig 1).

**Timing of antenatal care visits.** The median time of presenting for the first ANC care was 5 months (IQR: 4–6). Over the time period, there is strong evidence of association between timing of first ANC visit by survey year (P<0.001). The proportion of women with less than four ANC visits varied by survey year and month of first ANC visit. In general, the proportion of women with less than four ANC visits was the highest in 2010 while the least was observed in 2004.

The characteristics of women with at least four ANC visits regardless of the timing of the first ANC visit are shown in Table 4. A total of 12738 women had at least 4 ANC visits regardless of timing of their first visit. Of these, 9353 (74%) started ANC after the fourth month of pregnancy. Between 2004 and 2016, we observed a decreasing trend in the proportion of women with who attended their first ANC visit after four months from 88% in 2004/5 to 61% in 2015/16. There was increasing trend in proportion of women with late attendance of ANC by parity (see Table 4). However, increasing level of education was associated with increasing trend in the proportion of women with early attendance of ANC from 21% amongst those with no education to 52% amongst those with tertiary education.

## Discussion

The primary aim of this study was to explore the social and demographic factors associated with early initiation of ANC4+ in women in Malawi between 2004 and 2016 using MDHS data

**Table 3. Bivariate and multivariate odds ratios for factors associated with attending at least four or more antenatal care visits prior to prior to six months of first antenatal care visit in Malawi, 2004–2016.**

| Characteristics (n = 26386) | Bivariate analysis | | Multivariate analysis | |
|---|---|---|---|---|
| | OR (95%CI) | P-value | OR (95%CI) | P-value |
| **Age group** | | | | |
| 15–19 | 1.00 | | 1.00 | |
| 20–24 | 1.06 (0.90–1.26) | 0.49 | 1.27 (1.05–1.53) | 0.01 |
| 25–29 | 1.05 (0.88–1.25)0.59 | 0.22 | 1.44 (1.15–1.81) | 0.002 |
| 30–34 | 1.02 (0.85–1.23) | 0.82 | 1.49 (1.15–1.93) | 0.003 |
| 35–39 | 1.04 (0.85–1.28) | 0.69 | 1.64 (1.22–2.20) | 0.001 |
| 40–44 | 0.90 (0.68–1.18) | 0.44 | 1.51 (1.05–2.16) | 0.02 |
| 45–49 | 1.06 (0.70–1.61) | 0.77 | 1.91 (1.18–3.09) | 0.008 |
| **Year** | | | | |
| 2004/5 | 1.00 | | 1.00 | |
| 2010 | 1.16 (1.02–1.33) | 0.025 | 1.15 (1.01–1.32) | 0.04 |
| 2015/16 | 2.15 (1.89–2.45) | <0.001 | 2.03 (1.78–2.32) | <0.001 |
| **Region** | | | | |
| North | 1.00 | | | |
| Centre | 0.96 (0.84–1.11) | 0.56 | | |
| South | 0.91 (0.79–1.04) | 0.18 | | |
| **Number of children ever born** | | | | |
| 1 | 1.00 | | 1.00 | |
| 2–3 | 0.80 (0.71–0.90) | <0.001 | 0.74 (0.63–0.86) | <0.001 |
| 4–5 | 0.74 (0.65–0.85) | <0.001 | 0.65 (0.53–0.80) | <0.001 |
| 6+ | 0.69 (0.59–0.80) | <0.001 | 0.61 (0.47–0.79) | <0.001 |
| **Education level** | | | | |
| None | 1.00 | | 1.00 | |
| Primary | 1.23 (1.08–1.41) | 0.003 | 1.10 (0.95–1.28) | 0.19 |
| Secondary | 1.72 (1.46–2.03) | <0.000 | 1.24 (1.02–1.51) | 0.032 |
| Tertiary | 4.95 (3.20–7.66) | <0.001 | 2.64 (1.65–4.22) | <0.001 |
| **Wealth index quintile** | | | | |
| Poorest | 1.00 | | 1.00 | |
| Poorer | 1.12 (0.97–1.30) | 0.11 | 1.14 (0.98–1.31) | 0.09 |
| Middle | 1.13 (0.98–1.31) | 0.10 | 1.15 (1.00–1.34) | 0.06 |
| Richer | 1.19 (1.02–1.38) | 0.023 | 1.16 (0.99–1.35) | 0.06 |
| Richest | 1.60 (1.37–1.86) | <0.001 | 1.32 (1.12–1.58) | <0.001 |
| **Residence** | | | | |
| Urban | 1.00 | | | |
| Rural | 0.72 (0.63–0.83) | <0.001 | | |
| **Sources of antenatal care knowledge** | | | | |
| **Frequency of listening to radio** | | | | |
| Less than once a week | 1.00 | | | |
| At least once a week | 1.00 (0.91–1.10) | 1.00 | | |
| **Frequency of watching television** | | | | |
| Less than once a week | 1.00 | | | |
| At least once a week | 1.53 (1.31–1.79) | <0.001 | | |
| **Barriers to access antenatal care** | | | | |
| **Permission to visit health services** | | | | |
| No problem | 1.00 | | | |

*(Continued)*

**Table 3.** (Continued)

| Characteristics (n = 26386) | Bivariate analysis | | Multivariate analysis | |
|---|---|---|---|---|
| | OR (95%CI) | P-value | OR (95%CI) | P-value |
| Big problem | 1.09 (0.95–1.25) | 0.23 | | |
| **Money to pay for health services** | | | | |
| No problem | 1.00 | | | |
| Big problem | 0.86 (0.78–0.94) | 0.001 | | |
| **Distance to health facilities** | | | | |
| No problem | 1.00 | | | |
| Big problem | 0.90 (0.82–0.99) | 0.028 | | |
| **Presence of companion** | | | | |
| No problem | 1.00 | | | |
| Big problem | 0.99 (0.89–1.09) | 0.80 | | |
| **No drugs at health facility** | | | | |
| No problem | 1.00 | | | |
| Big problem | 1.25 (1.14–1.37) | <0.001 | | |
| **No female provider** | | | | |
| No problem | 1.00 | | | |
| Big problem | 0.98 (0.87–1.09) | 0.67 | | |
| **Marital status** | | | | |
| Never married | 1.00 | | | |
| Married | 1.11 (0.85–1.45) | 0.45 | | |
| Widowed | 0.98 (0.62–1.55) | 0.94 | | |
| Divorced | 1.03 (0.76–1.40) | 0.85 | | |

OR = weighted odds ratios of attending at least four antenatal care (ANC) visits prior to six months of first ANC visit.

95%CI = 95% Confidence Interval.

from three nationally representative surveys. Most studies exploring the factors associated with ANC uptake have not taken into account the month of the first ANC visit in calculating the distributions of women with ANC4+ visits. To our knowledge, this is the first study in Malawi that has analysed the likelihood of a pregnant woman having early initiation of ANC4 + visits. Whilst attendance of ANC4+ has often been a focus of ANC service use research in SSA, both early attendance for the first ANC visit and undertaking the recommended number of visits per-pregnancy are important for pregnancy outcomes. Early initiation of ANC allows healthcare workers to improve both maternal and perinatal outcomes by undertaking screening and tests that are more efficacious in early pregnancy, including accurate gestational dating or screening for maternal anaemia [9]. Identification of these complications early allows for the appropriate management for the length of pregnancy to improve outcomes. It is equally important that women also undertake visits for the length of their pregnancy, not only so indicated screening and treatment can continue, but to prepare women for birth and potential complications of delivery, which can lead to increased likelihood of women seeking facility delivery [26].

The results of this study provide vital insight into how coverage of ANC4+ changed during this time period and may highlight potential barriers that could be faced whilst rolling-out the updated WHO eight 'contact' ANC model, through identification of which women are at risk of attending ANC too late and at an insufficient frequency. However, it should be noted that whilst early initiation of ANC is significantly associated with attendance of ANC4+ and ANC8

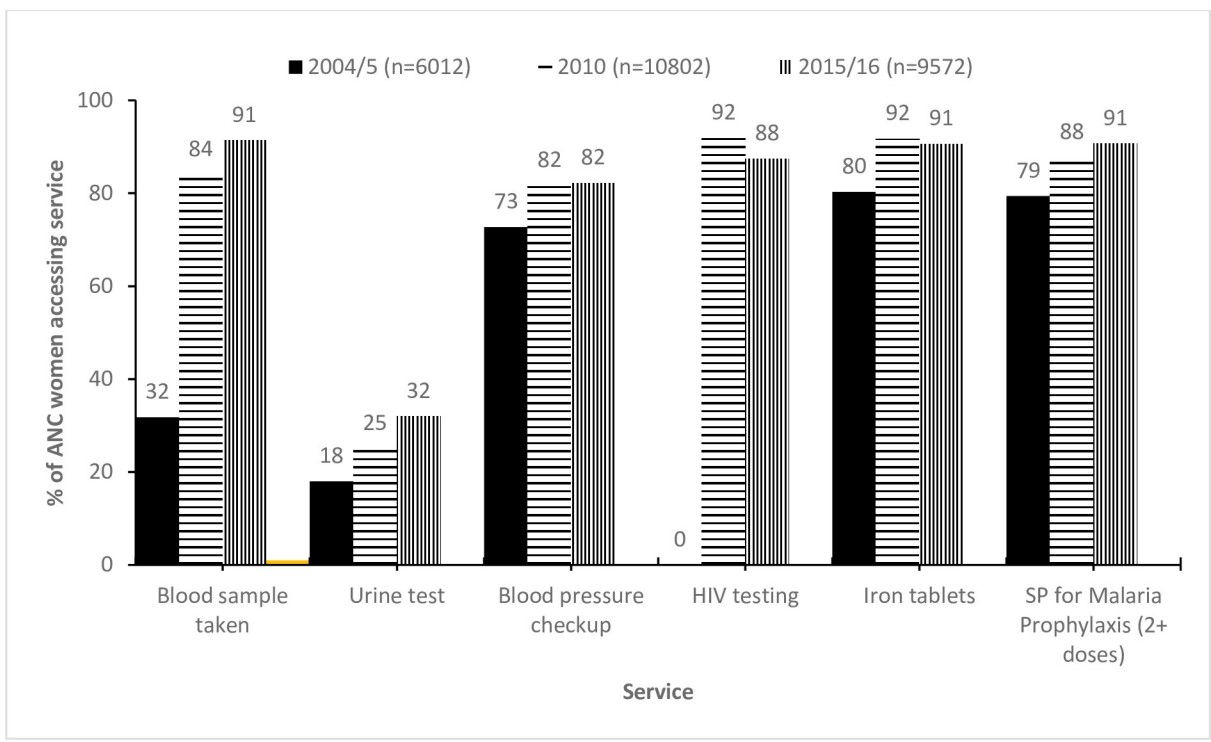

**Fig 1. Services received by women that attended antenatal care in Malawi between 2004 and 2016.** SP = Sulfadoxine-Pyrimethamine, % = weighted percentage.

+ in other LMIC settings [8], other socio-demographic determinants of ANC8+ in Malawi may not be consistent with those found in this study due to increased time commitment and possible associated costs of attending an increased number of visits. Additionally, the focus of this study is limited to individual-level socio-demographic factors and their influence on early initiation of ANC4+. As we outline below, we are unable to fully explore the effect of other key determinants of health care seeking such as quality of care on ANC attendance as this is not suitably captured within the datasets we used for our analysis.

Significant predictors of early initiation of ANC4+ attendance, determined through our analysis and discussed below, included maternal factors (number of children ever born, age, marital status) and socio-economic factors (wealth quintile and education status). The positive relationship between higher wealth and education status and increased likelihood of early initiation of ANC4+ highlight considerable inequalities present in ANC attendance in the population. Our results demonstrate that women are significantly more likely to have initiated ANC4 + early in 2015 than in 2004 (21% of women in 2004 vs 37% of women in 2015). Similarly, women who received care were more likely to receive key interventions during their ANC as we found substantial increases in the coverage of essential interventions between 2004 and 2016 including undertaking full blood counts, performing urine dipsticks to detect bacteremia and/or pre-eclampsia, malaria prophylaxis and HIV testing suggesting an overall improvement in quality of ANC services. Whilst this is promising, only 37% of women surveyed in 2015 had initiated ANC4+ early. With the sub-optimal number of women attending the ANC4+, the implementation of the at least eight ANC visits in settings like Malawi may not be feasible as echoed by authors of studies analyzing coverage of ANC in other LIC settings [8].

**Table 4.  Distribution of women with at least four antenatal care visits by the socio-demographic characteristics of the women in Malawi between 2004 and 2016.**

| Characteristics | ANC attendance for women with ANC4+ | | | |
| --- | --- | --- | --- | --- |
| | Early | | Late | |
| | n | % | n | % |
| **Total** | 3385 | 26.0 | 9353 | 74.0 |
| **Age group** | | | | |
| 15–19 | 327 | 26.4 | 888 | 73.6 |
| 20–24 | 1046 | 26.5 | 2867 | 73.5 |
| 25–29 | 870 | 25.9 | 2371 | 74.1 |
| 30–34 | 581 | 26.4 | 1604 | 73.6 |
| 35–39 | 373 | 25.5 | 1035 | 74.5 |
| 40–44 | 133 | 22.8 | 444 | 77.2 |
| 45–49 | 55 | 26.2 | 144 | 73.8 |
| **Year** | | | | |
| 2004/5 | 405 | 12.1 | 2957 | 87.9 |
| 2010 | 1081 | 22.5 | 3507 | 77.5 |
| 2015/16 | 1899 | 39.1 | 2889 | 60.9 |
| **Region** | | | | |
| North | 674 | 30.2 | 1523 | 69.8 |
| Centre | 1158 | 25.0 | 3396 | 75.0 |
| South | 1553 | 25.5 | 4434 | 74.5 |
| **Number of children ever born** | | | | |
| 1 | 834 | 27.5 | 2151 | 72.5 |
| 2–3 | 1255 | 26.9 | 3297 | 73.1 |
| 4–5 | 771 | 25.3 | 2233 | 74.7 |
| 6+ | 525 | 23.1 | 1672 | 76.9 |
| **Education level** | | | | |
| None | 417 | 21.3 | 1541 | 78.7 |
| Primary | 2224 | 25.9 | 6177 | 74.1 |
| Secondary | 645 | 28.1 | 1545 | 71.9 |
| Tertiary | 99 | 52.0 | 90 | 48.0 |
| **Wealth index quintile** | | | | |
| Poorest | 691 | 27.7 | 1807 | 72.3 |
| Poorer | 678 | 23.9 | 2139 | 76.1 |
| Middle | 708 | 25.1 | 1988 | 74.9 |
| Richer | 637 | 25.8 | 1803 | 74.2 |
| Richest | 671 | 28.0 | 1616 | 72.0 |
| **Residence** | | | | |
| Urban | 527 | 28.0 | 1268 | 72.0 |
| Rural | 2858 | 25.7 | 8085 | 74.3 |
| **Sources of antenatal care knowledge** | | | | |
| **Frequency of listening to radio** | | | | |
| Less than once a week | 1805 | 28.1 | 4486 | 71.9 |
| At least once a week | 1580 | 24.0 | 4867 | 76.0 |
| **Frequency of watching television** | | | | |
| Less than once a week | 2982 | 25.5 | 8515 | 74.5 |
| At least once a week | 403 | 30.8 | 838 | 69.2 |
| **Barriers to access antenatal care** | | | | |

(*Continued*)

**Table 4.** (Continued)

| Characteristics | ANC attendance for women with ANC4+ | | | |
| --- | --- | --- | --- | --- |
| | Early | | Late | |
| | n | % | n | % |
| **Permission to visit health services** | | | | |
| No problem | 2934 | 25.6 | 8247 | 74.4 |
| Big problem | 451 | 29.0 | 1106 | 71.0 |
| **Money to pay for health services** | | | | |
| No problem | 1716 | 28.4 | 4158 | 71.6 |
| Big problem | 1669 | 24.1 | 5195 | 75.9 |
| **Distance to health facilities** | | | | |
| No problem | 1530 | 27.1 | 3914 | 72.9 |
| Big problem | 1855 | 25.2 | 5439 | 74.8 |
| **Presence of companion** | | | | |
| No problem | 2454 | 26.0 | 6720 | 74.0 |
| Big problem | 931 | 26.0 | 2633 | 74.0 |
| **No drugs at health facility** | | | | |
| No problem | 1532 | 21.9 | 5277 | 78.1 |
| Big problem | 1853 | 30.5 | 4076 | 69.5 |
| **No female provider** | | | | |
| No problem | 2717 | 25.9 | 7556 | 74.1 |
| Big problem | 668 | 26.5 | 1797 | 73.5 |
| **Marital status** | | | | |
| Never married | 102 | 25.8 | 274 | 74.2 |
| Married | 2903 | 25.8 | 8079 | 74.2 |
| Widowed | 46 | 22.2 | 170 | 77.8 |
| Divorced | 334 | 28.5 | 830 | 71.5 |

% = weighted percentage.

We have reported a number of key determinants of early initiation of ANC4+ in this study. Increasing number of children ever born was associated with a reduced likelihood of ANC4+, with those women who have delivered more than 5 children being the least likely to undergo early initiation of ANC4+ visits. These findings are largely consistent with literature both in Malawi [13] and across SSA [27] and other LIC with high MMR [13]. It is plausible that women who have given birth and accessed these services before, possibly multiple times, are less likely to seek care as they feel that they are equipped with sufficient knowledge to proceed without formal maternity care. Additionally, the quality of maternity care women receive in Malawi remains variable, with some women reporting extensive stock outs and reception of treatment that lacked dignity or respect [28]. Although we infer quality of ANC service, we did not have data on quality of ANC services as such data were not collected by the MDHS. Women who have experienced poor quality may be less likely to seek care again, something that may happen with increasing frequency as number of children ever born increases. The relationship between increasing number of children ever born and reduced access to services is not unique to ANC but is present across other essential maternal and child health services both in Malawi and a number of countries with higher fertility rates [28]. The link between higher order births and increased risk of maternal and perinatal mortality is well described in the literature and it is possible that the inverse relationship between number of children ever born and service-use may be a contributing factor in this phenomenon [29].

Interestingly, whilst increasing number of children ever born was found to be associated with a reduced likelihood of early initiation of ANC4+, the opposite appears true regarding age, as women aged between 45 and 49 in our study population were nearly twice as likely to attend ANC4+ earlier than the youngest mothers after adjusting for number of children ever born and other factors. Whilst this association has been supported by findings from a number of studies exploring coverage of ANC, it has not been universally reported in the literature [27]. Older age does seem to affect care-seeking for other maternity services with older nulliparous women more likely to access facility delivery than their younger counterparts in a number of SSA regions [30].

Factors pertaining to a woman's socio-economic status, namely wealth quintile and level of education were found to be associated with early initiation of ANC4+ in this study. Women in the highest wealth quintile and those who had undertaken tertiary education were more likely to have initiated ANC early than their counterparts. The associations between both education and wealth level and maternal care seeking are similarly well documented in SSA and is not limited to access to ANC [13,27,31]. Despite women from lower wealth quintiles being least likely demonstrate early initiation of ANC4+ services provided through both public and Christian mission (CHAM) facilities in Malawi have been exempt from user-fees, and therefore free at the point of use, since 2004. This suggests that out-of-pocket payments provided to healthcare workers when accessing services is unlikely to explain the relationship between wealth level and the primary outcome. This relationship however could be explained by other costs associated with accessing services such as transportation. Initial exemption of user-fees following the initialization of service-level agreement between the Malawian Ministry of Health and CHAM facilities lead to increased utilization of maternity services in CHAM facilities [32]. However, in Tanzania cost was found to be a barrier for accessing ANC services [33].

Our study does have several limitations. Whilst the combined outcome variable capturing early attendance and complete attendance of the FANC model may be useful in determining who is more likely to engage successfully with ANC services, we were unable to assess the quality of these visits which may have impacted on a woman's propensity to seek further care. Additionally, the results for any variable captured in the MDHS survey at the time of administration may differ from the result at the time of last pregnancy and birth (i.e., changes in wealth or education over time). Similarly, time lag between administration and last birth may make survey responses vulnerable to recall bias and, as with all survey data, data captured in the MDHS is self-reported and subject to reporting bias, such as social desirability bias. We also recognize that the important factors which might influence ANC such as hypertension, diabetes and previous HIV status prior to the current pregnancy were not captured in the MDHS hence these were not included in this analysis. Finally, although HIV status being captured in the MDHS, there is no information on the timing of the HIV status to the pregnancy.

## Conclusion

In conclusion, whilst coverage of early initiation of ANC4+ and key ANC interventions in Malawi have increased significantly since 2004, there remains inequality in determinants of access to early initiation of ANC4+. Key socio-economic factors (education and and wealth) continue to impact women's likelihood of accessing these vital services and in 2016, just over a third of women surveyed were undertaking the recommended number of visits under the FANC model with the first ANC visit initiated early. Ensuring that all women are able to engage with ANC services at the appropriate point in their pregnancy and at the correct frequency will require consideration of the impact of inequality on how women engage with ANC services.

## Supporting information

**S1 File.**
(ZIP)

## Author Contributions

**Conceptualization:** Wingston Felix Ng'ambi, Joseph H. Collins, Tim Colbourn, Andrew Phillips, Timothy B. Hallett.

**Data curation:** Wingston Felix Ng'ambi.

**Formal analysis:** Wingston Felix Ng'ambi.

**Methodology:** Wingston Felix Ng'ambi, Tim Colbourn, Tara Mangal, Andrew Phillips, Timothy B. Hallett.

**Project administration:** Joseph Mfutso-Bengo, Paul Revill.

**Software:** Wingston Felix Ng'ambi.

**Validation:** Wingston Felix Ng'ambi, Andrew Phillips, Timothy B. Hallett.

**Visualization:** Wingston Felix Ng'ambi, Timothy B. Hallett.

**Writing – original draft:** Wingston Felix Ng'ambi, Joseph H. Collins.

**Writing – review & editing:** Wingston Felix Ng'ambi, Joseph H. Collins, Tim Colbourn, Tara Mangal, Andrew Phillips, Fannie Kachale, Joseph Mfutso-Bengo, Paul Revill, Timothy B. Hallett.

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
