## [Decision Letter · Decision Letter 0]

7 Apr 2021

PONE-D-21-03728

Socio-demographic factors associated with early antenatal care visits among pregnant women in Malawi: 2004-2016

PLOS ONE

Dear Dr. Ng'ambi,

Thank you for submitting your manuscript to PLOS ONE. After careful consideration, we feel that it has merit but does not fully meet PLOS ONE’s publication criteria as it currently stands. Therefore, we invite you to submit a revised version of the manuscript that addresses the points raised during the review process.

We look forward to receiving your revised manuscript.

Kind regards,

Orvalho Augusto, MD, MPH

Academic Editor

PLOS ONE

Additional Editor Comments:

This manuscript covers a relevant topic for reduction of maternal and neonatal mortality and morbidity in Malawi. The authors took 3 publicly available community surveys datasets from Malawi and conducted a secondary analysis to determine what variables are associated with the earlier initiation of antenatal care (ANC) and attendance of four or more ANC visits. They use a new and useful way of combining these two outcomes together by counting 4 or more only if the woman did the first ANC visit in the first trimester of gestation and attended by a skilled provider. This way it avoids overestimation of the true proportion of women with 4 or ANC visits since women initiating ANC visits after the first trimester tend to have multiple visits due to complications. They define a binary outcome ANC4+ according to this definition. For the analysis, they appended the 3 community surveys datasets into one dataset with 26,386 records and conducted univariate and multiple logistic regressions.

I must comment the authors for the really well writing.

Issues:

1. The authors state that they used survey weights as they were offered in the dataset. This would be fine if the analysis of each survey was done separately. And then do some combination of those estimates. But here, apparently the analysis was done as if we had weights of just one survey. This is problematic. Please explain what was done to the weights prior to their use into the models.

2. As one of the reviewers comment below the main outcome here is a combination of the two other outcomes (earlier initiation of antenatal care, and attendance of four or more ANC visits). These captures different goals and the second outcome has the problem of change in policy over time.

3. Line 131 put space between primary and outcome.

4. Line 138. Stata is not an acronym. So do not write STATA write please Stata. See Official Stata documentation.

5. In the limitations or in discussion in general please point out that there was a change in the requirement of minimal ANC visits over the course of these surveys.

Journal Requirements:

"Funding for this project was provided to the University of York to implement the Thanzi la Onse

 (TLO) Programme by the Research Council of the United Kingdom (RCUK). During the study period, WN, JC,

TC, AP, TM, JMB, PR and TBH worked for this RCUK funded project."

Reviewers' comments:

Reviewer's Responses to Questions

**Comments to the Author**

1. Is the manuscript technically sound, and do the data support the conclusions?

Reviewer #1: Yes

Reviewer #2: Yes

Reviewer #3: Yes

2. Has the statistical analysis been performed appropriately and rigorously? 

Reviewer #1: Yes

Reviewer #2: Yes

Reviewer #3: No

3. Have the authors made all data underlying the findings in their manuscript fully available?

Reviewer #1: Yes

Reviewer #2: Yes

Reviewer #3: Yes

4. Is the manuscript presented in an intelligible fashion and written in standard English?

Reviewer #1: Yes

Reviewer #2: Yes

Reviewer #3: Yes

5. Review Comments to the Author

Reviewer #1: Well written research - from justification, to methods and conclusion. I would like to suggest the following:

1. The underlying premise/assumption should be made clear and mentioned; women when instructed by health professionals to have 4 visits that their performance will be the same/projected to be the same when they are recommended to have 8 ANC visits. This may not be necessarily true.

2. It is important to address the socio-demographic factors in the discussion as "markers" for health professionals to pick up and potentially provide a "person-centered approach" in health management to ensure that maximum health services offered given the socio-economic constraints

3. "watching TV`' is a proxy to wealth and do not find it useful to highlight it as a finding for any use. Health care providers will be less likely to ask client if she watched TV as part of clinical encounter and similarly this will not be considered for health policy action. Would suggest to drop this variable.

4. Question: Are the health services really "free"? Often, in countries where this is stated, there are other expenses paid during a health visit. I would suggest not to rule this out and consider transport costs as the only limiting factor.

5. The authors touched on briefly the "Quality of Care" and "respectful care" - these elements are often very much related to setting, low quality (lower levels of care with potentially stock outs) and respect issues with low paid/unsupervised/unregualted health professionals in rural/low socio-economic settings. This needs further elaboration in the discussion and needs to mentioned that this was not part of data

6. SSA acronym missing in list of abbreviations

7. Conclusion section should not only address policy makers/program managers. Please see point 2.

Otherwise congratulations to research team for this manuscript!

Reviewer #2: The paper is relevant and touches on an important aspect of maternal and child health. The following are my comments:

General comment: Correct any spelling errors, eg line 234/235 should read “of a pregnant woman” and not “of a pregnant women”.

Methods: provide a description of the study area.

Reviewer #3: The authors modelled socio-demographic factors associated with early initiation (within four months of pregnancy) of first ANC contact and attending at least four ANC visits (ANC4+) in Malawi using data collected in 2004, 2010, and 2016 Malawi Demographic and Health Survey (MDHS) health surveys. These ANC data were collected before 2016 WHO revised recommendations of increasing the number of antenatal care (ANC) visits per pregnancy from four to eight. The outcome variable was binary on attending at least 4 ANC visits, with a first visit occurring during or before 4 months gestation. A binary regression was used to ascertain association with several purported factors. Predictors were included in the model based on their univariate association having a likelihood ratio test at less than P< significant level. The paper is well written and researched and add base knowledge on the uptake of modern ANC care. However, I have several concerns about data description and statistical elements.

a) There is clarity of the numbers of women interviewed for the ain survey and the women who and a pregnancy/birth in the last two years of the surveys, which is the same used here. Please could you add a column showing the number of women who were pregnant in the last two years versus the number of women interviewed for the respective main surveys as an indication of external validity.

b) The sample weights in the respective surveys were valid and benchmarked to a survey. Once the data are combined, you can not use the original weights since the circumstances have changed, the weight will need to reduce. Thus, please could adjusting the weights in the combined data set.

c) Two outcomes are combined: early ANC initiation and number of ANC visits. I think the two serve different though similar purposes in ANC care; the first helps to early problems detection and managing them during the pregnancy time; the second for measuring and monitoring pregnant woman contact with skilled health personnel. So would rather you analyse three outcomes: early ANC visit, ANC4, and combined.

d) What using a cut pint of 8 ANC, will Malawi have already passed the new 2016 recommendations? Or rather at the rate, when will Malawi achieve this? Then how will you advise the MoH in Malawi?

6. PLOS authors have the option to publish the peer review history of their article (what does this mean?). If published, this will include your full peer review and any attached files.

Reviewer #1: **Yes: **Wisal Mustafa Hassan Ahmed

Reviewer #2: No

Reviewer #3: **Yes: **Samuel Manda

---

## [Author Response · Author response to Decision Letter 0]

4 Jun 2021

Additional Editor Comments:

This manuscript covers a relevant topic for reduction of maternal and neonatal mortality and morbidity in Malawi. The authors took 3 publicly available community surveys datasets from Malawi and conducted a secondary analysis to determine what variables are associated with the earlier initiation of antenatal care (ANC) and attendance of four or more ANC visits. They use a new and useful way of combining these two outcomes together by counting 4 or more only if the woman did the first ANC visit in the first trimester of gestation and attended by a skilled provider. This way it avoids overestimation of the true proportion of women with 4 or ANC visits since women initiating ANC visits after the first trimester tend to have multiple visits due to complications. They define a binary outcome ANC4+ according to this definition. For the analysis, they appended the 3 community surveys datasets into one dataset with 26,386 records and conducted univariate and multiple logistic regressions.

I must comment the authors for the really well writing.

Issues:

1. The authors state that they used survey weights as they were offered in the dataset. This would be fine if the analysis of each survey was done separately. And then do some combination of those estimates. But here, apparently the analysis was done as if we had weights of just one survey. This is problematic. Please explain what was done to the weights prior to their use into the models.

RESPONSE: Thank you for your comment. As you mention above, the original weights were calculated separately for each survey round of the DHS and stored as the weighting variable V005. As instructed by the DHS program, this variable was divided by 1000000 to come up with the sample weight used in the analysis. Our analytical approach follows the standards recommended by the DHS program for DHS data analysis, which we have now referenced within the manuscript. 

We set the data to incorporate the weights using the STATA command: svyset [pw=wgt], psu(PSU). For regression models we used importance weights as implemented in this code for the final model: xi: logit anc i.age_gr i.region i.ceb i.edlev i.windex i.year i.radfreq i.mstatus[iw=wgt], or. It should also be noted that we stratified the analysis by year for all the tables except Table 3 and Table 4. In Table 3 we assessed the effect of the year on the outcome hence we did not stratify the analysis. In Table 4 we aimed to show the proportion of women presenting early or late amongst all the women with at least 4 ANC visits.

2. As one of the reviewers comment below the main outcome here is a combination of the two other outcomes (earlier initiation of antenatal care, and attendance of four or more ANC visits). These captures different goals and the second outcome has the problem of change in policy over time.

RESPONSE: Thank you for this comment. We agree that that early initiation of ANC and ‘complete’ ANC attendance are different yet interrelated outcomes when evaluating access to ANC services. Considering ANC4+ in isolation does not provide sufficient insight into the distribution of visits across the entire pregnancy. Similarly, whilst we believe early attendance of ANC1 is associated with ‘complete’ ANC attendance, using early attendance as the sole primary outcome does not capture the coverage of later, and equally important, visits during pregnancy.

Both early initiation and ‘complete’ attendance of the ANC scheduled are important, undoubtedly for different reasons, and therefore combining them into one primary outcome provides a more accurate representation of access to ANC in this setting. We have highlighted in our introduction why we feel it is important that women attend both early and complete ‘set’ of visits in lines (81-87). “ Ensuring both early access to ANC and ANC4+ for women in Malawi is important, as not only does ANC lead to improved maternal [10], newborn [14] and early childhood outcomes [15] but early initiation of ANC is positively associated with women attending both ANC4+ and attending at eight or more ANC visits in other settings [8]. “

Regarding the effect of policy change over time, our analysis uses data that was collected prior to the adoption of the 8-contact schedule in Malawi therefore we are only able to focus on understanding what access to ANC4 was prior to the policy change. We agree that understanding determinants of ANC8+ is obviously a very important and pertinent research question within Malawi but this is outside the remit of this paper (and the current availability of national data) and we feel that our analysis is still important in evaluating women’s access of services prior to the adoption of a new policy. 

3. Line 131 put space between primary and outcome.

RESPONSE: Thank you for highlighting we have now revised accordingly

4. Line 138. Stata is not an acronym. So do not write STATA write please Stata. See Official Stata documentation.

RESPONSE: Thank you for highlighting we have now revised accordingly 

5. In the limitations or in discussion in general please point out that there was a change in the requirement of minimal ANC visits over the course of these surveys.

 RESPONSE: Thank you for your comment. As highlighted above data collection for the 2015-2016 DHS occurred prior to the publication of the WHO recommendations for 8 ANC contacts in 2016. Therefore, these recommendations had not been adopted by the Malawian government during the data collection period. We have added a sentence to our introduction to clarify this: lines 99-100: “. We have undertaken an analysis of MDHS data which was collected prior to publication of the WHO’s 2016 guidelines and adoption of these guidelines by the Malawian government.”

Journal Requirements:

"Funding for this project was provided to the University of York to implement the Thanzi la Onse

 (TLO) Programme by the Research Council of the United Kingdom (RCUK). During the study period, WN, JC,

TC, AP, TM, JMB, PR and TBH worked for this RCUK funded project."

RESPONSE: Thank you. We have updated the declaration section as follows “

Declarations

We declare that there is no conflict of interest in publishing this paper. The authors would like to thank the MEASURE DHS for granting access to use the datasets for Malawi. Funding for the project was provided by UK Research and Innovation (UKRI) through the GCRF Thanzi la Onse (Health of All) research programme (MR/P028004/1). During the study period, WN, JC, TC, AP, TM, JMB, PR and TBH worked for on the project. The funder had no role in the study design, data collection and analysis, decision to publish, or the presentation of the manuscript.

” 

Reviewers' comments:

Reviewer's Responses to Questions

Comments to the Author

1. Is the manuscript technically sound, and do the data support the conclusions?

Reviewer #1: Yes

Reviewer #2: Yes

Reviewer #3: Yes

2. Has the statistical analysis been performed appropriately and rigorously? 

Reviewer #1: Yes

Reviewer #2: Yes

Reviewer #3: No

3. Have the authors made all data underlying the findings in their manuscript fully available?

Reviewer #1: Yes

Reviewer #2: Yes

Reviewer #3: Yes

4. Is the manuscript presented in an intelligible fashion and written in standard English?

Reviewer #1: Yes

Reviewer #2: Yes

Reviewer #3: Yes

5. Review Comments to the Author

Reviewer #1: Well written research - from justification, to methods and conclusion. I would like to suggest the following:

1. The underlying premise/assumption should be made clear and mentioned; women when instructed by health professionals to have 4 visits that their performance will be the same/projected to be the same when they are recommended to have 8 ANC visits. This may not be necessarily true.

RESPONSE: Thank you for your comment. We agree that there are a number of complex socio-demographic and health system factors which influence women’s attendance of a prescribed ‘set number’ of ANC contacts which may vary when changing the primary outcome from 4 contacts to 8.

As these data were collected before the initiation of 2016 ‘8 contacts’ WHO recommendation we are unable to determine if there are common determinants of attending ANC4+ and ANC8+ in Malawi. However, we are fairly confident that in other low- and middle-income country settings early initiation of ANC has a significant impact on a woman’s likelihood on attending 4+ or 8+ visits as referenced in the introduction of our paper (Jiwani et al. 2020- reference 8 in manuscript reference list). As our primary outcome incorporates timely initiation of ANC it may be reasonable to assume that the determinants of our primary outcome could also affect likelihood of attending 8+ visits, although this would require further study. 

We have added these clarifications to the second paragraph of our discussion, lines 261-271: 

“The results of this study provide vital insight into how coverage of ANC4+ changed during this time period and may highlight potential barriers that could be faced whilst rolling-out the updated WHO eight ‘contact’ ANC model, through identification of which women are at risk of attending ANC too late and at an insufficient frequency. However, it should be noted that whilst early initiation of ANC is significantly associated with attendance of ANC4+ and ANC8+ in other LMIC settings [8], other socio-demographic determinants of ANC8+ in Malawi may not be consistent with those found in this study due to increased time commitment and possible associated costs of attending an increased number of visits.” 

2. It is important to address the socio-demographic factors in the discussion as "markers" for health professionals to pick up and potentially provide a "person-centered approach" in health management to ensure that maximum health services offered given the socio-economic constraints

RESPONSE: Thank you. Whilst we agree in the need for consideration of socio-demographic factors in health management we feel that through our discussion we provide ample exploration of the importance of the variables that we have found to impact access to services. We feel that providing detailed direction on how to incorporate these factors into service management is outside of the remit of this paper. 

3. "watching TV`' is a proxy to wealth and do not find it useful to highlight it as a finding for any use. Health care providers will be less likely to ask client if she watched TV as part of clinical encounter and similarly this will not be considered for health policy action. Would suggest to drop this variable.

RESPONSE: We have removed this sentence from the abstract as agree it is not worth highlighting given it will not be considered directly for health policy action. As you mention “watching TV” is a proxy for wealth therefore we retain it in our analysis, also because studies conducted in similar settings included this variable under exposure to media (Rwabilimbo et al 2020, reference 32 of manuscript) and including such a variable therefore enhances comparability of the study findings under these settings. 

4. Question: Are the health services really "free"? Often, in countries where this is stated, there are other expenses paid during a health visit. I would suggest not to rule this out and consider transport costs as the only limiting factor.

RESPONSE: Thank you, we agree that in this, and other similar settings there is certainly a number of unmeasured costs associated with accessing health services. In Malawi, as we highlight in our discussion, health services are provided free of charge through the public sector but the patient pays their own transport to and from the health facility, we now have highlighted in-text that this may be a factor that affects decision to seek care: lines 340-343: “This suggests that out-of-pocket payments provided to healthcare workers when accessing services is unlikely to explain the relationship between wealth level and the primary outcome. This relationship however could be explained by other costs associated with accessing services such as transportation.”

Through our multivariate analysis, as seen in Table 3, we found no independent effect of women’s perception that ‘money to pay for health services’ or ‘distance to health facility’ being a ‘big problem’ when accessing services on our primary outcome. 

5. The authors touched on briefly the "Quality of Care" and "respectful care" - these elements are often very much related to setting, low quality (lower levels of care with potentially stock outs) and respect issues with low paid/unsupervised/unregualted health professionals in rural/low socio-economic settings. This needs further elaboration in the discussion and needs to mentioned that this was not part of data

RESPONSE: Thank you for your comment. The focus of our study is purely on individual level socio-demographic factors on early initiation of ANC4+ in Malawi and we have taken care to ensure this is made clear, for instance in the titling of our manuscript and through the description of our methods. We agree that there are many other factors, including actual and perceived quality of service provision, that will impact individual’s propensity to seek care and that they have not been explored in this paper. We have added to our discussion to clarify that we are unable to explore the effect of variables related to quality of care on care seeking and that these variables likely have a significant effect at line (256-258). 

“Additionally, the focus of this study is limited to individual-level socio-demographic factors and their influence on early initiation of ANC4+. As we outline below, we are unable to fully explore the effect of other key determinants of health care seeking such as quality of care on ANC attendance as this is not suitably captured within the datasets we used for our analysis.”

6. SSA acronym missing in list of abbreviations

RESPONSE: Thank you for highlighting this, we have now included SSA in the list of abbreviations.

7. Conclusion section should not only address policy makers/program managers. Please see point 2.

RESPONSE: Thank you. We agree that policy makers/program managers are not that only target audience for this paper and have amended our conclusion accordingly. 

Otherwise congratulations to research team for this manuscript!

RESPONSE: Thank you very much!

Reviewer #2: The paper is relevant and touches on an important aspect of maternal and child health. The following are my comments:

General comment: Correct any spelling errors, eg line 234/235 should read “of a pregnant woman” and not “of a pregnant women”.

RESPONSE: Thank you for highlighting this, we have made the correction and thoroughly checked through our manuscript for spelling errors.

Methods: provide a description of the study area.

RESPONSE: The study involves all the 28 districts of Malawi. More details are provided in the manuscript from line 108 to 121.

Reviewer #3: The authors modelled socio-demographic factors associated with early initiation (within four months of pregnancy) of first ANC contact and attending at least four ANC visits (ANC4+) in Malawi using data collected in 2004, 2010, and 2016 Malawi Demographic and Health Survey (MDHS) health surveys. These ANC data were collected before 2016 WHO revised recommendations of increasing the number of antenatal care (ANC) visits per pregnancy from four to eight. The outcome variable was binary on attending at least 4 ANC visits, with a first visit occurring during or before 4 months gestation. A binary regression was used to ascertain association with several purported factors. Predictors were included in the model based on their univariate association having a likelihood ratio test at less than P< significant level. The paper is well written and researched and add base knowledge on the uptake of modern ANC care. However, I have several concerns about data description and statistical elements.

a) There is clarity of the numbers of women interviewed for the aim survey and the women who and a pregnancy/birth in the last two years of the surveys, which is the same used here. Please could you add a column showing the number of women who were pregnant in the last two years versus the number of women interviewed for the respective main surveys as an indication of external validity.

RESPONSE: Thank you very much for the compliment. The MDHS only captures detailed information on the most recent birth (in the 24 months prior to the survey) and no other births were captured. We therefore feel we would not be able to accurately generate a column in table one which detailed the number of women who were pregnant in the survey year as requested above. Through the sampling frame employed by the MDHS we feel that the sample from which our data has been taken provides a representative sample of the population of Malawi and suitably demonstrates the external validity of our methodology. 

b) The sample weights in the respective surveys were valid and benchmarked to a survey. Once the data are combined, you can not use the original weights since the circumstances have changed, the weight will need to reduce. Thus, please could adjusting the weights in the combined data set.

RESPONSE: The weights were specific for each survey round and our analysis did not change. All we did as shown in Table 1, we stratified the analysis by survey year and pooled the results as well. As for the regression analysis, we used importance weights and included the variable year as an explanatory variable. 

c) Two outcomes are combined: early ANC initiation and number of ANC visits. I think the two serve different though similar purposes in ANC care; the first helps to early problems detection and managing them during the pregnancy time; the second for measuring and monitoring pregnant woman contact with skilled health personnel. So would rather you analyse three outcomes: early ANC visit, ANC4, and combined.

RESPONSE: Thank you for your comment. We agree that both early initiation of ANC and ‘complete’ attendance of ANC (i.e. 4 visits) are important outcomes when evaluating access to ANC services in any setting. We also feel that the combination of these outcomes into one primary outcome, as is the methodology of our analysis in this manuscript, provides a novel analysis of DHS data which we believe provides a better indication of ‘complete’ coverage of ANC as its captures these two key outcomes as interrelated and equally important in coverage of services. 

As we have highlighted above in response to the editors comments we feel that previous analyses which use four or more ANC visits as the primary outcome have only evaluated the proportion of women attending four ANC visits regardless of timing of the first ANC visit- we felt this was inadequate within our analysis for the reasons described above. 

Therefore, we do not feel that separating the primary outcome into its composite parts (early initiation, ANC4) as you suggest here would add to our analysis as our primary aim was to ascertain the proportion of women for which both of these outcomes are true. We note that kind words of the editor in their description of our primary outcome as “a new and useful way of combining these two outcomes together by counting 4 or more only if the woman did the first ANC visit in the first trimester of gestation and attended by a skilled provider. This way it avoids overestimation of the true proportion of women with 4 or ANC visits since women initiating ANC visits after the first trimester tend to have multiple visits due to complications”. In addition please note that in table 4 you are able to see the proportion of women who attend ANC4+ stratified by early or late initiation of ANC1. 

d) What using a cut pint of 8 ANC, will Malawi have already passed the new 2016 recommendations? Or rather at the rate, when will Malawi achieve this? Then how will you advise the MoH in Malawi?

RESPONSE: Whilst we agree that this is certainly a vital research question in its own right we feel that this is outside of the remit of the analysis within this current manuscript and we would be unable to draw accurate conclusions or predictions about this using the MDHS data, which was collected before the policy of 8 ANC visits started. Our analysis does show that the ‘complete’ ANC attendance was lower than recommended by the FANC policy in place in Malawi at the time and therefore we do highlight in the conclusion section that this may mean that Malawi is less likely to achieve high coverage of ANC8 as coverage of ANC4 was low prior to policy change.

6. PLOS authors have the option to publish the peer review history of their article (what does this mean?). If published, this will include your full peer review and any attached files.

Do you want your identity to be public for this peer review? For information about this choice, including consent withdrawal, please see our Privacy Policy.

Reviewer #1: Yes: Wisal Mustafa Hassan Ahmed

Reviewer #2: No

Reviewer #3: Yes: Samuel Manda

---

## [Decision Letter · Decision Letter 1]

13 Jul 2021

PONE-D-21-03728R1

Socio-demographic factors associated with early antenatal care visits among pregnant women in Malawi: 2004-2016

PLOS ONE

Dear Dr. Ng'ambi,

Thank you for submitting your manuscript to PLOS ONE. After careful consideration, we feel that it has merit but does not fully meet PLOS ONE’s publication criteria as it currently stands. Therefore, we invite you to submit a revised version of the manuscript that addresses the points raised during the review process.

We look forward to receiving your revised manuscript.

Kind regards,

Orvalho Augusto, MD, MPH

Academic Editor

PLOS ONE

Journal Requirements:

Additional Editor Comments:

Major issue:

The authors seem to have misunderstood the message about the weights. The current description of the weights use is for just a single survey procedure. When there is a combination of surveys the weights must be corrected. Please do see the comments from the reviewer bellow. And please add the details of such weighting procedure in the statistical analysis section.

Minor issues:

Abstract

- Put the the ORs and 95%CI for the factors you mention in the results

- There still “STATA” instead of “Stata” in the abstract.

Line 91 “..” corect to be “.”

Methods

- There is nowhere in this manuscript a description of what women are included in the analysis. Line 105 says it is a secondary analysis of the women’s questionnaire data. Not many readers of PlosONE will know what is this questionnaire. Please state briefly what women are included in the survey (women with a pregnancy in the last 5 years prior to the survey, for example). Such statement should be added to the abstract as well.

Results

- Table 1 and 2 are using weights? Please add footnote explaining that.

Reviewers' comments:

Reviewer's Responses to Questions

**Comments to the Author**

1. If the authors have adequately addressed your comments raised in a previous round of review and you feel that this manuscript is now acceptable for publication, you may indicate that here to bypass the “Comments to the Author” section, enter your conflict of interest statement in the “Confidential to Editor” section, and submit your "Accept" recommendation.

Reviewer #1: All comments have been addressed

Reviewer #3: (No Response)

2. Is the manuscript technically sound, and do the data support the conclusions?

Reviewer #1: Yes

Reviewer #3: Yes

3. Has the statistical analysis been performed appropriately and rigorously? 

Reviewer #1: I Don't Know

Reviewer #3: Yes

4. Have the authors made all data underlying the findings in their manuscript fully available?

Reviewer #1: Yes

Reviewer #3: Yes

5. Is the manuscript presented in an intelligible fashion and written in standard English?

Reviewer #1: Yes

Reviewer #3: Yes

6. Review Comments to the Author

Reviewer #1: (No Response)

Reviewer #3: Thanks to the authors for attending to the revision of their paper. I am largely satisfied but there a few clarifications which they may want to consider, and hopefully, make the paper better.

a) On the question of the number of women having a most recent birth (in the 24 months before the survey); in my previous comments, I was pointing to having a column indicating the overall number of women aged 15-49 interviewed for the main survey vs the women having a most recent birth (in the 24 months before the survey). For example, in there a total of 11,698 women (15-49) were interviewed for the main questionnaire vs 6012 (51%%). Also, the captions of Tables 1 and 2 may be misleading at first sight because it talks of women interviewed, rather than women with a recent birth (24 months before the survey). Thus, if one puts a column on Table 1 indicating the number of women from which this sample came, e.g. 2,407 women in the age group 15-19 vs 602 analyzed here, etc.

b) I applaud that the authors in accounting for the design of the surveys by way of weighted analyses. And in most surveys, designs weights are combined with missing data and non-response. However, when analyzing combined data of multiple MDHS datasets, individual survey weights would need to be adjusted correctly to represent a case in the combined data sets. See for example.

Alexander, C. H. (2002). Still Rolling: Leslie Kish's ‘Rolling Samples’ and The American Community Survey. Survey Methodology, 28:1, 35-41

Friedman, E.M., Jang, D. & Williams T.V. (2002) Combined estimates from four quarterly survey data sets. Proceedings of the American Statistical Association Joint Statistical Meetings - Section on Survey Research Methods, pp.1064-1069. Alexandria, VA: American Statistical Association.

c) On the question of 8+ ANC visits, one could use it as sensitivity analysis.

7. PLOS authors have the option to publish the peer review history of their article (what does this mean?). If published, this will include your full peer review and any attached files.

Reviewer #1: **Yes: **Wisal Mustafa Hassan Ahmed

Reviewer #3: No

---

## [Author Response · Author response to Decision Letter 1]

25 Aug 2021

Response to Editor Comments 

1.) “The authors seem to have misunderstood the message about the weights. The current description of the weights use is for just a single survey procedure. When there is a combination of surveys the weights must be corrected. Please do see the comments from the reviewer bellow. And please add the details of such weighting procedure in the statistical analysis section.”

Thank you for highlighting this. We have amended the weighting procedure as directed and made changes to the manuscript accordingly. Please see response to reviewer three for a detailed description of our actions. 

2.) “Put the ORs and 95%CI for the factors you mention in the results”

Thank you, we have now added the ORs and 95% CI for all results that are reported in the abstract. Changes can be seen at lines 34-44, as indicated by highlighting. In addition, we have added that the results for the primary outcome into the abstract as well as seen at line 35.

3.) “There still “STATA” instead of “Stata” in the abstract.”

Thank you for highlighting. This has been amended as show at line 25.

4.) “Line 91 “..” correct to be “.””

Thank you for highlighting. This has been amended as show at line 99.

5.) “Methods - There is nowhere in this manuscript a description of what women are included in the analysis. Line 105 says it is a secondary analysis of the women’s questionnaire data. Not many readers of PlosONE will know what is this questionnaire. Please state briefly what women are included in the survey (women with a pregnancy in the last 5 years prior to the survey, for example). Such statement should be added to the abstract as well.”

We have now amended the manuscript to include a description of the survey sample in both the abstract, at line 25-27, and in the main text at line 115 - 122. 

6.) “Results - Table 1 and 2 are using weights? Please add footnote explaining that.”

Thank you for highlighting. We have now added footnotes to all tables and figures to show that weights were used in calculating proportions/percentages and odds ratios. 

Response to Reviewer Comments 

a) On the question of the number of women having a most recent birth (in the 24 months before the survey); in my previous comments, I was pointing to having a column indicating the overall number of women aged 15-49 interviewed for the main survey vs the women having a most recent birth (in the 24 months before the survey). For example, in there a total of 11,698 women (15-49) were interviewed for the main questionnaire vs 6012 (51%%). Also, the captions of Tables 1 and 2 may be misleading at first sight because it talks of women interviewed, rather than women with a recent birth (24 months before the survey). Thus, if one puts a column on Table 1 indicating the number of women from which this sample came, e.g. 2,407 women in the age group 15-19 vs 602 analyzed here, etc.

Thank you for the clarification of your previous comment. We feel that addition of the column may not be of any value as such women to be included in the other column did not have a recent birth of whom their antenatal care is being assessed in terms of healthcare seeking behaviour.

Additionally, we have updated the titles of table 1 and 2 to more accurately reflect their contents, in this case that they refer exclusively to women who delivered in the past five years of the survey. 

b) I applaud that the authors in accounting for the design of the surveys by way of weighted analyses. And in most surveys, designs weights are combined with missing data and non-response. However, when analyzing combined data of multiple MDHS datasets, individual survey weights would need to be adjusted correctly to represent a case in the combined data sets. See for example.

Thank you very much for the direction regarding the survey weights and for providing additional sources that we were able to use to strengthen our analysis. We have now employed an equal weights approach as heighted in Friedman et al (2002). We have provided a description of our methods at line 149 and added Friedman et al’s paper to the reference list at number 25.

Following this weight approach we found that ‘frequency of watching television’ which was significant in the previous analysis is no longer significant factor in the presence of other variables. Further, different percentages as shown by Tables and Figures also changed. Please refer to the Tables and Figures for more details on the change.

c) On the question of 8+ ANC visits, one could use it as sensitivity analysis.

As for 8+ ANC, we had 513 (2% of 26,234) women who had at least 8 ANC visits. We did not perform the sensitivity analysis since the numbers are small and a future analysis should examine this after when the recommendations have been in place for longer.

---

## [Decision Letter · Decision Letter 2]

25 Jan 2022

Socio-demographic factors associated with early antenatal care visits among pregnant women in Malawi: 2004-2016

PONE-D-21-03728R2

Dear Dr. Ng'ambi,

We’re pleased to inform you that your manuscript has been judged scientifically suitable for publication and will be formally accepted for publication once it meets all outstanding technical requirements.

Kind regards,

Orvalho Augusto, MD, MPH

Academic Editor

PLOS ONE

Additional Editor Comments (optional):

Reviewers' comments:

Reviewer's Responses to Questions

**Comments to the Author**

1. If the authors have adequately addressed your comments raised in a previous round of review and you feel that this manuscript is now acceptable for publication, you may indicate that here to bypass the “Comments to the Author” section, enter your conflict of interest statement in the “Confidential to Editor” section, and submit your "Accept" recommendation.

Reviewer #1: All comments have been addressed

Reviewer #3: All comments have been addressed

2. Is the manuscript technically sound, and do the data support the conclusions?

Reviewer #1: Yes

Reviewer #3: Yes

3. Has the statistical analysis been performed appropriately and rigorously? 

Reviewer #1: I Don't Know

Reviewer #3: Yes

4. Have the authors made all data underlying the findings in their manuscript fully available?

Reviewer #1: Yes

Reviewer #3: Yes

5. Is the manuscript presented in an intelligible fashion and written in standard English?

Reviewer #1: Yes

Reviewer #3: Yes

6. Review Comments to the Author

Reviewer #1: (No Response)

Reviewer #3: (No Response)

7. PLOS authors have the option to publish the peer review history of their article (what does this mean?). If published, this will include your full peer review and any attached files.

Reviewer #1: **Yes: **WISAL MUSTAFA HASSAN AHMED

Reviewer #3: No

---

## [Editor Report · Acceptance letter]

31 Jan 2022

PONE-D-21-03728R2 

Socio-demographic factors associated with early antenatal care visits among pregnant women in Malawi: 2004-2016 

Dear Dr. Ng'ambi:

I'm pleased to inform you that your manuscript has been deemed suitable for publication in PLOS ONE. Congratulations! Your manuscript is now with our production department. 

Kind regards, 

on behalf of

Dr. Orvalho Augusto 

Academic Editor

PLOS ONE